# Synthetic enzyme-substrate tethering obviates the Tolloid-ECM interaction during *Drosophila* BMP gradient formation

**Jennifer Winstanley[1], Annick Sawala[1†], Clair Baldock[2]\*, Hilary L Ashe[1]\***

[1]Faculty of Life Sciences, University of Manchester, Manchester, United Kingdom; [2]Wellcome Trust Centre for Cell-Matrix Research, Faculty of Life Sciences, University of Manchester, Manchester, United Kingdom

**Abstract** Members of the Tolloid family of metalloproteinases liberate BMPs from inhibitory complexes to regulate BMP gradient formation during embryonic dorsal-ventral axis patterning. Here, we determine mechanistically how Tolloid activity is regulated by its non-catalytic CUB domains in the *Drosophila* embryo. We show that Tolloid, via its N-terminal CUB domains, interacts with Collagen IV, which enhances Tolloid activity towards its substrate Sog, and facilitates Tsg-dependent stimulation of cleavage. In contrast, the two most C-terminal Tld CUB domains mediate Sog interaction to facilitate its processing as, based on our structural data, Tolloid curvature positions bound Sog in proximity to the protease domain. Having ascribed functions to the Tolloid non-catalytic domains, we recapitulate embryonic BMP gradient formation in their absence, by artificially tethering the Tld protease domain to Sog. Our studies highlight how the bipartite function of Tolloid CUB domains, in substrate and ECM interactions, fine-tune protease activity to a particular developmental context.

**\*For correspondence:** clair.baldock@manchester.ac.uk (CB); hilary.ashe@manchester.ac.uk (HLA)

**Present address:** †Division of Physiology and Metabolism, Medical Research Council National Institute for Medical Research, London, United Kingdom

**Competing interests:** The authors declare that no competing interests exist.

## Introduction

Bone Morphogenetic Proteins (BMPs) represent a conserved family of secreted signalling proteins that regulate diverse processes during development and homeostasis (*Wu and Hill, 2009*). Multiple tiers of regulation within the BMP signalling pathway exist, including extracellular modulators that form inhibitory complexes with BMPs to prevent ligand interaction (*Wharton and Serpe, 2013*). These extracellular BMP inhibitors are crucial during dorsal-ventral axis patterning of the early embryo, where they act to redistribute broadly expressed BMP ligands to form a gradient of BMP activity.

The mechanism of BMP gradient formation has been extensively studied in the early *Drosophila* embryo, where a heterodimer of the BMP ligands, Dpp and Scw, acts as the most potent BMP signalling species (*Shimmi et al., 2005*). In dorsolateral regions, an inhibitory complex is formed, with the Dpp-Scw heterodimer bound by the extracellular antagonists Sog and Tsg (*Wharton and Serpe, 2013*), aided by the scaffold protein Collagen IV (*Wang et al., 2008*; *Sawala et al., 2012*; *Wharton and Serpe, 2013*). In this inhibitory complex, the Dpp-Scw heterodimer is unable to bind to receptors or Collagen IV, but is free to diffuse dorsally. The secreted Tld metalloproteinase cleaves Sog to release active BMP ligand, however in dorsolateral regions due to the high concentration of Sog, the ligand dimer is rebound. Thus it is through a cycle of complex formation, diffusion and cleavage that the ligands accumulate at the dorsal midline where, upon cleavage by Tld, Dpp-Scw is free to signal (*Wharton and Serpe, 2013*). A second phase of gradient formation involving intracellular positive feedback then refines this initial broad Dpp-Scw gradient into a peak of BMP receptor activation at the dorsal midline (*Wang and Ferguson, 2005*; *Gavin-Smyth et al., 2013*).

**eLife digest** The body of an animal is a highly organised structure of tissues and organs that contain cells with specialised roles. To achieve this level of organisation, it is important that the cells in the embryo know their location and receive the correct instructions on how to develop, when to divide or move. Many animals are roughly symmetrical about an imaginary line that runs from their head to their tail; a developing embryo can provide its cells with information about their position along this head-to-tail axis and the axis that runs from its front to its back.

Setting up the front-to-back axis in the embryo involves a family of proteins called the bone morphogenetic proteins (or BMPs). These proteins can bind to other proteins that act as signals to provide instructions to cells. However, many of the BMPs are unable to perform this job because they are trapped by inhibitory molecules that bind to them instead.

Enzymes belonging to the Tolloid family can break down these inhibitors to release the BMPs. Together, the inhibitors and Tolloid enzymes create a gradient of BMP activity across the embryo. The side of the embryo with the highest levels of active BMPs sets the position of the back of the body, while the opposite side—which has the lowest levels of active BMPs—becomes the front. However, it is not clear how Tolloid is controlled to create the BMP gradient.

Different parts of the Tolloid enzyme have different roles; one portion of the enzyme breaks down the inhibitory molecules, and there are also several so-called 'non-catalytic domains'. Winstanley et al. used a combination of approaches to study how Tolloid is controlled in fruit fly embryos. The experiments show that two non-catalytic domains at one end of Tolloid help the enzyme to bind to the inhibitory molecules. At the other end of the Tolloid enzyme, another non-catalytic domain can bind to a structural protein called Collagen IV. This enhances the ability of the enzyme to break down the inhibitory molecules and release the BMPs.

These findings reveal how Tolloid's non-catalytic domains can fine-tune the activity of this enzyme to create the gradient of BMP activity that is needed to set the front-to-back direction in animal embryos. Future studies will focus on identifying other proteins that bind to the non-catalytic domains of Tolloid in order to further control its activity during development.

Enzymes within the Tld family have a well-characterised domain structure in which the N-terminal protease domain is followed by a series of non-catalytic CUB (Complement-Uegf-BMP1) and EGF (Epidermal Growth Factor-like) domains (*Muir and Greenspan, 2011*) (*Figure 1A*). Structural analysis of mammalian Tolloids has revealed that both mammalian Tolloid (mTld) and Tolloid-like-1 (Tll-1) form dimers in which the most C-terminal EGF and CUB domains act in a substrate exclusion mechanism to restrict enzyme activity (*Berry et al., 2009, 2010*). In contrast, the highly active, shorter BMP1 protease is monomeric (*Berry et al., 2009*). Although *Drosophila* Tld possesses a similar domain structure to mTld, previous studies suggest that removing the three most C-terminal domains, to mimic a shorter BMP1-like form, results in a loss of activity (*Canty et al., 2006*). Therefore, a substrate exclusion mechanism may not exist for *Drosophila* Tld. In addition, despite single amino acid mutations in four out of the five CUB domains resulting in some degree of ventralisation of the *Drosophila* embryo, the requirement for these domains in *Drosophila* Tld function is unclear (*Ferguson and Anderson, 1992*; *Childs and O'Connor, 1994*; *Finelli et al., 1994*).

Here we combine biophysical, biochemical and genetic approaches to determine how the domain structure of Tld acts to regulate its enzymatic activity. We propose that the role of *Drosophila* Tld CUB domains can be segregated; the most C-terminal domains are necessary for Sog interaction whilst the N-terminal domains manage a novel interaction with Collagen IV that acts to enhance Tld activity. In addition, the Tld-Collagen IV interaction is important for the Tsg enhancement of Sog processing by Tld. We also demonstrate that the need for Tld CUB domains can be bypassed by engineering the Tld–Sog interaction in a highly efficient manner.

## Results

### The monomeric structure of *Drosophila* Tld

We investigated the structure of *Drosophila* Tld purified from tissue culture cells using Small Angle X-ray Scattering (SAXS). *Drosophila* Tld has a maximal particle dimension ($D_{max}$) of 17.7 nm and a radius

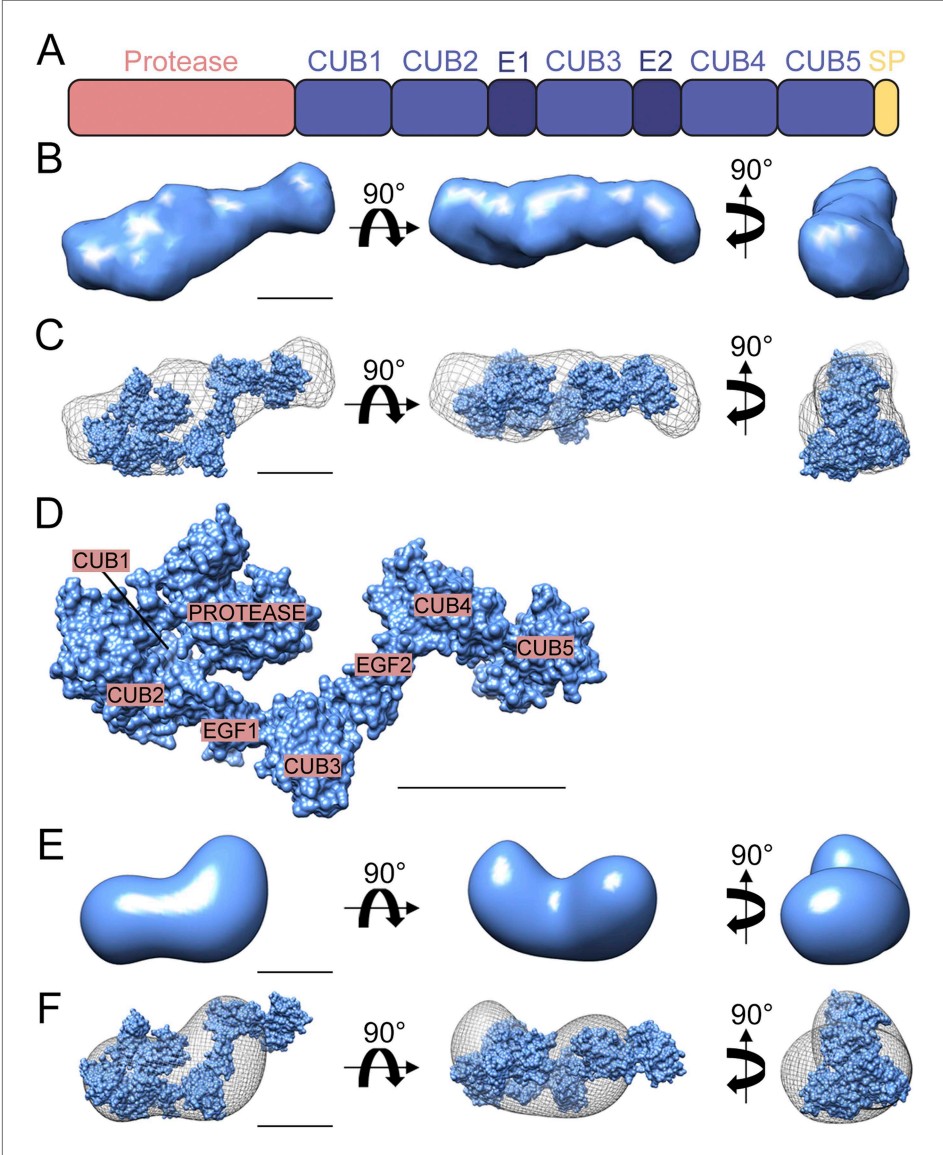

**Figure 1**. Structure of *Drosophila* Tld. (**A**) Cartoon of *Drosophila* Tld domain organisation. E = EGF-like domain. SP = specific peptide. (**B**) Low resolution *ab initio* model generated by DAMMIN and DAMAVER from solution SAXS data shown in three orthogonal orientations. (**C**) Overlay of the *ab initio* model shown in (**B**) displayed as chicken-wire with a representative rigid body model generated by SASREF. (**D**) The rigid body model shown in (**C**) with Tld domains labelled. (**E**) 3D reconstruction of the *Drosophila* Tld monomer by single particle analysis electron microscopy shown in three orthogonal orientations. (**F**) The rigid body model was fitted into the EM volume shown in (**E**), here displayed as chicken-wire. All scale bars = 5 nm. See also *Figure 1—figure supplements 1 and 2*.

The following figure supplements are available for figure 1:

**Figure supplement 1**. *Drosophila* Tld SAXS data analysis.

**Figure supplement 2**. *Drosophila* Tld TEM data.

of gyration (Rg) of 4.6 nm (*Figure 1—figure supplement 1*). *Ab inito* modelling of the most probable shape of *Drosophila* Tld gives an elongated monomer with the approximate dimensions 18 × 6.5 × 6.8 nm (*Figure 1B*). Further detail was extrapolated from the SAXS data on the location of the CUB and EGF domains of Tld by rigid body modelling (RBM). The model retains similar characteristics to the SAXS bead model with the approximate dimensions of 16.6 × 7 × 9.2 nm. An overlay of the *ab inito* model

and rigid body model (*Figure 1C*) shows that Tld displays an elongated conformation with protein curvature around the C-terminal tail (*Figure 1D*). This brings the most C-terminal CUB domains into close proximity to the N-terminal protease domain. Single particle analysis TEM demonstrates that *Drosophila* Tld is a monomer, with the overall shape of the 3D reconstruction in agreement with the SAXS model (*Figure 1E*, *Figure 1—figure supplement 2*). In addition to being relatively elongated, the curvature of the EM model mimics that seen in the RBM (*Figure 1F*). However, it is evident that the TEM model is shorter; this suggests flexibility of the most C-terminal regions that is likely to result in loss of information on this region during the averaging procedure.

## CUB4 and CUB5 are necessary for substrate interaction

We investigated the role of individual CUB domains, with particular interest in the most C-terminal CUB domains, CUB4 and CUB5, due to their position within the rigid body model structure. Each domain was independently deleted in full-length Tld and the ability of these variants, as well as the isolated metalloprotease (MP) domain, to cleave Sog substrate was tested (*Figure 2A,B*). Cleavage of Sog by Tld relies on the presence of Dpp (*Marques et al., 1997*) (*Figure 2—figure supplement 1*), as such Dpp was added to all in vitro assays. Only ΔCUB2 retained the ability to cleave Sog, although the level of cleavage by ΔCUB2 is lower than full-length Tld (*Figure 2C*). Additionally, whilst increasing levels of Tsg enhances Tld cleavage of the Sog 50 and 33 kDa fragments, especially the 33 kDa fragment as described previously (*Shimmi and O'Connor, 2003*), ΔCUB2 mediated cleavage is not enhanced (*Figure 2C*). Therefore, while CUB2 is not required for Tld activity, its presence is necessary for Tsg enhancement of cleavage.

In order to test whether the loss of activity of these Tld variants is due to an effect on substrate interaction, the ability of each Tld protein to bind to Sog was assessed using immunoprecipitation. To facilitate detection of the Sog–Tld interaction, for all Sog binding experiments described here we used catalytically inactive versions of Tld and the various derivatives tested, by introducing the previously described E94A mutation into the protease active site (*Garrigue-Antar et al., 2001*). Analysis of Sog–Tld interactions reveals that, similar to the isolated protease domain, the CUB4 and CUB5 deletions are unable to interact with Sog, suggesting that these domains mediate Sog interaction (*Figure 2D*). To determine whether the loss of activity for the ΔCUB4 and ΔCUB5 forms is due to a specific requirement for both CUB4 and CUB5 domains in Sog interaction rather than an issue of protein length, full-length variants of Tld were generated with CUB4 or CUB5 domain duplications at the C-terminus. However, these enzymes are also inactive, demonstrating a specific requirement for the CUB4–CUB5 domain pair at the C-terminus of Tld (*Figure 2—figure supplement 2*). For the other single CUB deletions, ΔCUB2 interacts with Sog to a similar extent as wildtype Tld even though its activity is reduced and, despite being inactive, the ΔCUB1 and ΔCUB3 forms can bind Sog, albeit at reduced levels (*Figure 2D*). These data suggest that a correct protein conformation is important for Tld activity, or that CUB1 and CUB3 mediate a different protein interaction that is important for Sog binding (see later).

## Previously identified Tld point mutants affect Sog binding

Four single amino acid mutations situated within CUB4 or CUB5 have previously been isolated from genetic screens and described in terms of the cuticle phenotype and the strength of genetic interactions with *dpp* mutations (*Ferguson and Anderson, 1992*; *Childs and O'Connor, 1994*; *Finelli et al., 1994*) (*Figure 3A*), yet the molecular basis for the decrease in Tld activity is unclear. Therefore, based on our data that CUB4 and CUB5 mediate Sog interaction, we tested whether the molecular defect associated with these mutant Tld proteins is reduced binding to Sog. First we extended the phenotypic analysis for the *tld^E839K* mutation, by determining the effect of this mutation on the expression of target genes responding to different thresholds of Dpp activity. Visualisation of the expression patterns of the Dpp target genes *Race* and *u-shaped* (*ush*) in *tld^E839K* homozygous mutant embryos reveals that expression of the peak threshold gene *Race* is lost and expression of the lower threshold gene *ush* is narrower than in wild-type embryos (*Figure 3B*), suggesting shallow gradient formation, consistent with the previous characterisation of this allele as weak.

Identification of the positions of the four mutated residues within the individual CUB domain structures reveals that all lie on the surface of the protein (*Figure 3C*) and therefore can by hypothesised to be important for substrate binding. To directly test the effect of each mutation on Sog binding and cleavage activity, we engineered these mutations into Tld-HA and expressed the mutant

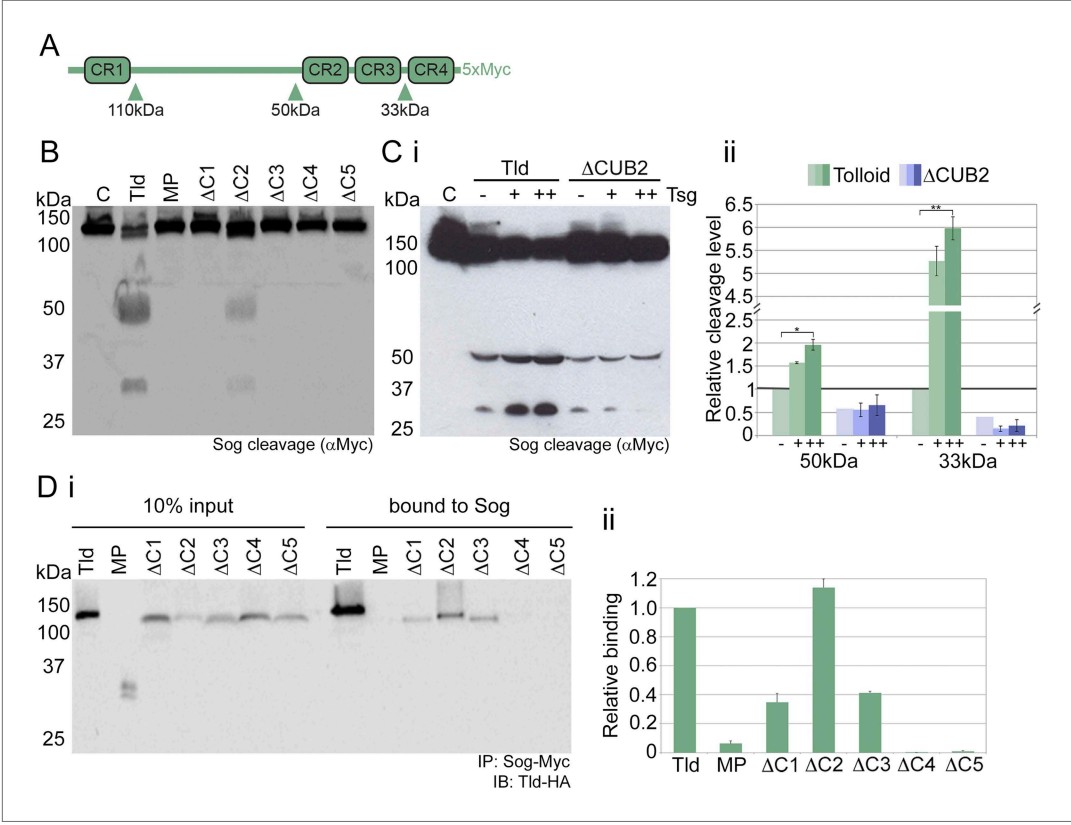

**Figure 2**. Tld CUB4/5 domains mediate Sog binding. (**A**) Schematic of Sog showing the CR domains and the position of the Tld cleavage sites. The sizes of the C-terminal fragments liberated by cleavage, as detected by the C-terminal Myc tag, are indicated below the arrows. (**B**) Western blot (anti-Myc) of Sog cleavage assays carried out using normalised amounts of the Tld deletion proteins indicated, all in the presence of Dpp and absence of Tsg. C = control assay with no Tld added. MP = the isolated metalloproteinase domain. (**Ci**) A representative Western blot (anti-Myc) showing Sog cleavage assays carried out with normalised levels of Tld and ΔCUB2 in the presence of Dpp and either in the absence (−), or with increasing amounts (+, ++) of Tsg. (**Cii**) Graph shows the amount of cleavage measured as the amount of the 50 kDa and 33 kDa fragments quantified relative to cleavage calculated for wildtype Tld in the absence of Tsg, based on three different experiments including the Western blot shown in **i**. Error bars show SEM, n = 3, *p < 0.005, **p < 0.001. (**Di**) Western blot (anti-HA) showing 10% input of catalytically inactive forms of Tld-HA proteins and the amount bound to Sog-Myc by immunoprecipitation. (**Dii**) Quantitation of the level of binding of each Tld protein to Sog, relative to binding of full-length Tld. Error bars are SEM, n = 4. See also *Figure 2—figure supplements 1 and 2*.

The following figure supplements are available for figure 2:

**Figure supplement 1**. Sog cleavage by Tld is dependent on Dpp.

**Figure supplement 2**. Tld CUB4-CUB5 domain pair is specifically required at the C-terminus.

proteins. Each mutant shows reduced binding to Sog (*Figure 3D*) and a loss of cleavage activity when tested over a short reaction time (*Figure 3Ei*), although upon a longer incubation time varying levels of weaker cleavage activity can be detected for all of the mutant Tld proteins tested (*Figure 3Eii*). Together, these data provide molecular understanding of how these mutations impair Tld activity, and provide in vivo support for the importance of CUB4 and CUB5 in mediating the Tld–Sog interaction.

## Collagen IV binds to Tld CUB1-3 domains to enhance cleavage activity

In a pilot mass spectrometry screen for Tld binding proteins we identified Collagen IV, an extracellular matrix protein that we have previously shown to be required for Dpp gradient formation

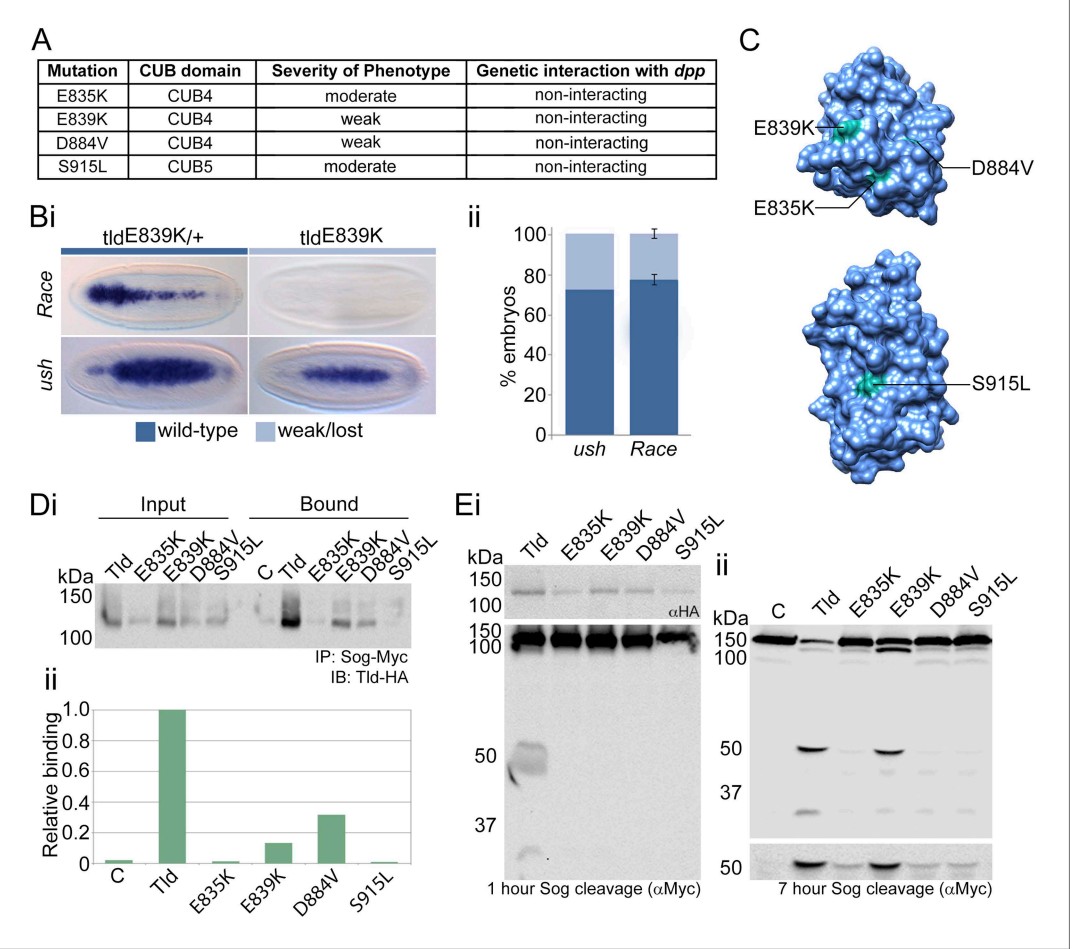

**Figure 3**. Altered Sog binding of Tld point mutants. (**A**) Table summarising previous experimental data regarding the CUB4 and CUB5 mutations and the severity of the resulting phenotypes (**Childs and O'Connor, 1994**; **Finelli et al., 1994**). (**Bi**) Dorsal views of stage 6 wildtype and $tld^{E839K}$ homozygous embryos stained by RNA in situ hybridisation for the Dpp target genes *Race* and *u*-shaped. Phenotypes were quantified in **Bii** as the percentage of embryos with either wildtype or lost/narrow expression for *Race* and *ush*. (**C**) Models of Tld CUB4 and CUB5 with the single amino acid mutations highlighted in green. (**Di**) Western blot with anti-HA showing the input levels of the catalytically inactive Tld-HA proteins tested and the amounts bound to Sog-Myc in an immunoprecipitation experiment. (**Dii**) Graph shows the level of binding to Sog relative to that for full-length Tld. (**E**) Western blot (top, anti-HA) showing the levels of Tld proteins carrying the indicated single amino acid changes. Western blots (anti-Myc) show 1 hr (**Ei**) or 7 hr (**Eii**) Sog cleavage assays using the Tld proteins shown. Below the blot on the right is a longer exposure of the 50 kDa cleavage fragment.

(**Wang et al., 2008**). To both validate a direct interaction between Tld and Collagen IV, and determine which domains in Tld are responsible for this interaction, we tested the ability of full-length Tld and the individual CUB domain deletion variants to bind to the NC1 domain of the Viking (Vkg) Collagen IV protein fused to GST. The Collagen IV NC1 domain is a highly conserved globular domain that has been shown to bind Dpp and Sog previously (**Wang et al., 2008**; **Sawala et al., 2012**). Tld binds to Collagen IV via its N-terminal CUB domains, with deletion of CUB1, CUB2 or CUB3 reducing binding to <25% that of full-length (**Figure 4A**). However, removal of the Sog interacting domains, CUB4 and CUB5, has little effect on the Tld-Collagen IV interaction. No interaction with the control GST protein was observed for any of the Tld-HA proteins (data not shown). To test the effect of Collagen IV on Tld and Sog interaction, their binding was investigated in the presence of an increased amount of Collagen IV. Addition of the NC1 domains from both *Drosophila* Collagen IV proteins, Vkg and Dcg1, increases the Sog–Tld interaction (**Figure 4B**).

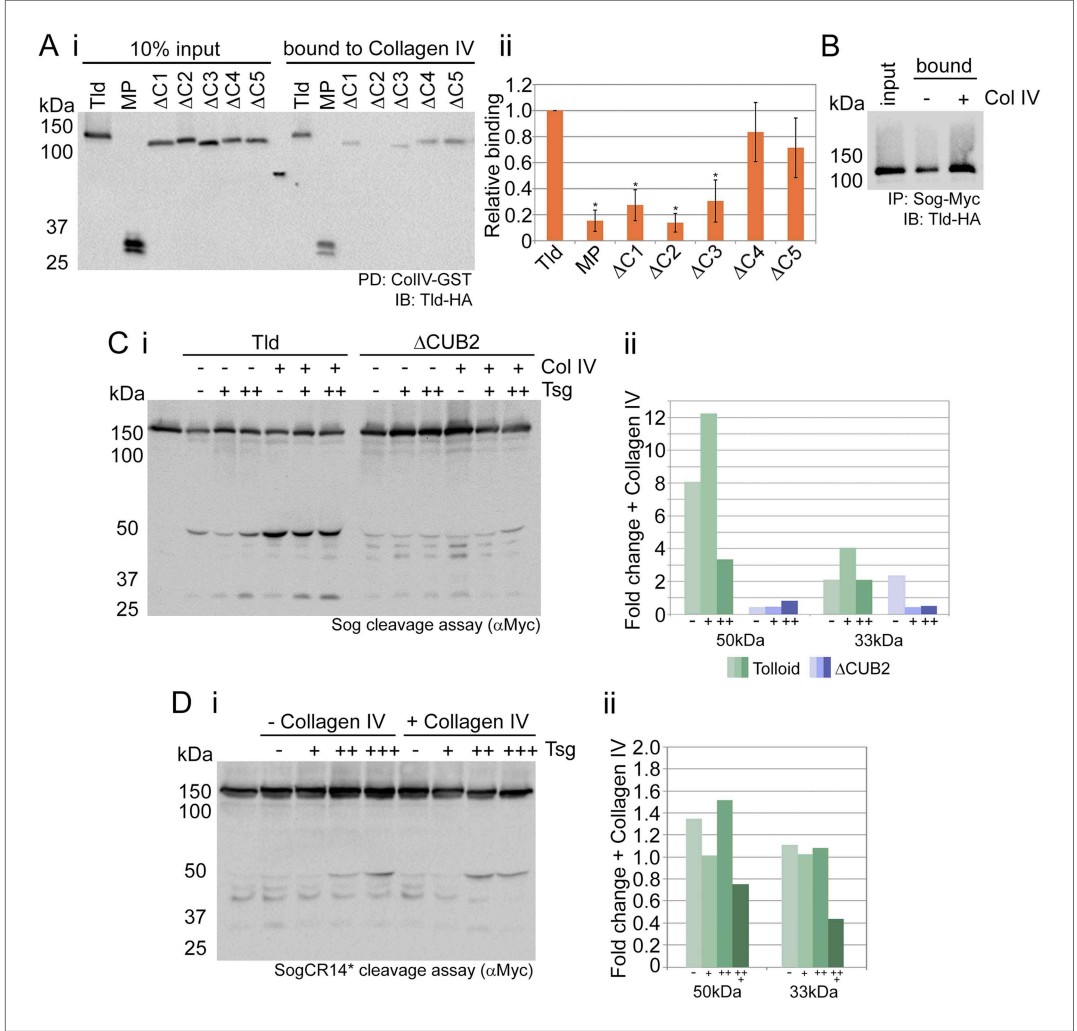

**Figure 4**. Tld binds Collagen IV. (**Ai**) Western blot showing Tld-HA input proteins and the amount bound to GST-VkgNC1 in a pulldown experiment. (**Aii**) Graph shows the level of binding to Collagen IV plotted relative to binding of full-length Tld, based on data from three experiments. Error bars are SEM, n = 3, *p < 0.05. (**B**) Western blot (anti-HA) showing Tld input and the levels bound to Sog in the presence of Dpp, with or without (±) added Collagen IV. (**Ci**) Western blot (anti-Myc) of Sog cleavage by full-length Tld or the ΔCUB2 mutant quantified in (**Cii**). The x-axis labels (–,+,++) refer to the absence, or increasing amounts, of Tsg. Error bars are SEM, n = 3. (**Di**) Western blot (anti-Myc) of Tld cleavage of SogCR14*, with cleavage levels quantified in (**Dii**). Both cleavage assays were carried out over 3 hr, in the presence of Dpp, and in the absence or presence of increasing amounts of Tsg, with or without of Collagen IV, as indicated. Graphs show the fold-change in the relative levels of the 50 kDa and 33 kDa cleavage fragments released by Tld (green) or ΔCUB2 (blue), in the presence of Collagen IV relative to its absence. See also *Figure 4—figure supplement 1*.

The following figure supplement is available for figure 4:

**Figure supplement 1**. SogCR14* is inefficiently processed by Tld in vivo.

Next, the activity of Tld was compared in the presence and absence of Collagen IV. It is likely that soluble Collagen IV, or its cleavage fragments following matrix turnover, are present in the media from transfected cells along with the test proteins. Therefore, to limit Collagen IV levels, test proteins were obtained from transfected cells that were also treated with *vkg* RNAi. Despite this, we expect that the '–Collagen IV' condition will still contain some Collagen IV, and compare this low level to a '+Collagen IV' condition in which the Vkg and Dcg1 NC1 domains were added. The addition of Collagen IV enhances Tld cleavage at both the 50 and 33 kDa sites of Sog (*Figure 4C*), consistent with increased Sog–Tld

interaction in the presence of Collagen IV (*Figure 4B*). The addition of Tsg increases Sog cleavage, especially at the 33 kDa site in the presence of Collagen IV. However, the Collagen IV enhancement of Sog processing is reduced in the presence of high levels of Tsg (*Figure 4C*). In contrast, little enhancement at either the 50 or 33 kDa site is observed in the presence of Collagen IV when the ΔCUB2 mutant, that is defective in Collagen IV interaction, is tested (*Figure 4C*). As before (*Figure 2C*), Tsg fails to enhance Sog processing by the ΔCUB2 mutant (*Figure 4C*), suggesting that Tsg function may depend on a Tld-Collagen IV interaction. These data show that Tld binding to Collagen IV enhances the Sog-Tld interaction and Sog cleavage.

To extend these findings, we investigated whether the Collagen IV-Sog interaction is also important for Sog processing by Tld. To this end, we made use of a mutant form of Sog, SogCR14*, we have described previously that harbours mutations in the CR1 and CR4 domains of Sog, resulting in very weak binding to Collagen IV (*Sawala et al., 2012*). This mutant Sog is a much poorer substrate for Tld cleavage compared to wildtype Sog, and no enhancement of cleavage in the presence of added Collagen IV is observed (*Figure 4D*), contrary to that observed with wildtype Sog (*Figure 4C*). In addition, there is very little of the 33 kDa cleavage fragment, even when Tsg is added in increasing amounts (*Figure 4D*), suggesting that Tsg function requires a Sog-Collagen IV interaction, in addition to a Tld-Collagen IV interaction (*Figure 4C*). Consistent with SogCR14* being a poorer Tld substrate in vitro, analysis of embryos expressing either wildtype Sog or the SogCR14* double mutant under the control of the *sog* lateral stripe (ls) enhancer reveals that the SogCR14* mutant accumulates to a greater level than the wildtype Sog in vivo (*Figure 4—figure supplement 1*). Together these data show that binding of both Sog and Tld to Collagen IV is necessary for efficient Sog processing.

## Tld binding to Collagen IV is important in vivo

We hypothesised that the role of CUB2 and CUB3 in Collagen IV interaction can be used to explain classical mutations situated in these domains (*Figure 5A*). The surface position of these mutations in the CUB2 and CUB3 domains is shown in *Figure 5B*. The mutations were engineered into Tld-HA and we first tested for Collagen IV binding, which is reduced by at least two-fold in all CUB2/3 point mutants (*Figure 5C*). We next tested the effect of these point mutations on Sog interaction. All Tld mutants show reduced Sog interaction, with the same severity profile as that observed for the defects in Collagen IV binding (*Figure 5Di,iii*, cf with *Figure 5Cii*), consistent with the data presented in *Figure 4* that Collagen IV promotes the Tld–Sog interaction. To show that the effects on Sog interaction are an indirect effect of the Tld mutant proteins being unable to bind Collagen IV, which promotes Sog–Tld binding, we uncoupled the assay from Collagen IV, again using the mutant form of Sog that is defective in Collagen IV binding (*Sawala et al., 2012*). As predicted, Tld binding to this mutant form of Sog is weaker as the Collagen IV enhancement is lost (*Figure 5Dii*). However, the four mutant Tld proteins all show equivalent levels of binding to this form of Sog, as that observed for wildtype Tld (*Figure 5Dii,iii*). These data reveal that the CUB2/3 mutations do not affect Sog interaction per se, but rather disrupt the stimulatory effect observed through Collagen IV. In terms of cleavage activity, the four mutant Tld proteins show reduced cleavage of Sog, especially of the 33 kDa fragment (*Figure 5E*), consistent with their reduced ability to bind Sog and the requirement for Collagen IV for Tsg enhancement of processing at the 33 kDa site. Together, these data suggest that the molecular defect underlying the phenotypes associated with the CUB2/3 point mutations is a weak ability to bind Collagen IV.

## Efficiently engineering the Tld–Sog interaction bypasses the need for the non-catalytic domains

Based on the functions we have identified for the non-catalytic domains of Tld, we hypothesised that using an artificial method to bring together Sog and the Tld MP domain would rescue protease activity, bypassing both the need for substrate interaction domains and Collagen IV enhancement. To this end leucine zipper sequences (Z+ and Z−) (*Luan et al., 2006*) were attached to the C-terminus of both the isolated Tld MP domain and full-length Sog (*Figure 6A*). Immunoprecipitations confirmed the ability of the leucine zippers to induce enzyme-substrate interaction; SogZ+ specifically interacts with the corresponding MPZ− zipper but not the MPZ+ or unmodified MP domain controls (*Figure 6B*). To test whether the interaction induced by the leucine zippers is sufficient to rescue the activity of the isolated protease domain, cleavage assays were carried out in cell culture. Both full-length Tld and MPZ− cleave SogZ+ to yield two cleavage fragments at 57 kDa and 40 kDa (*Figure 6C*), equivalent to the 50 and 33 kDa

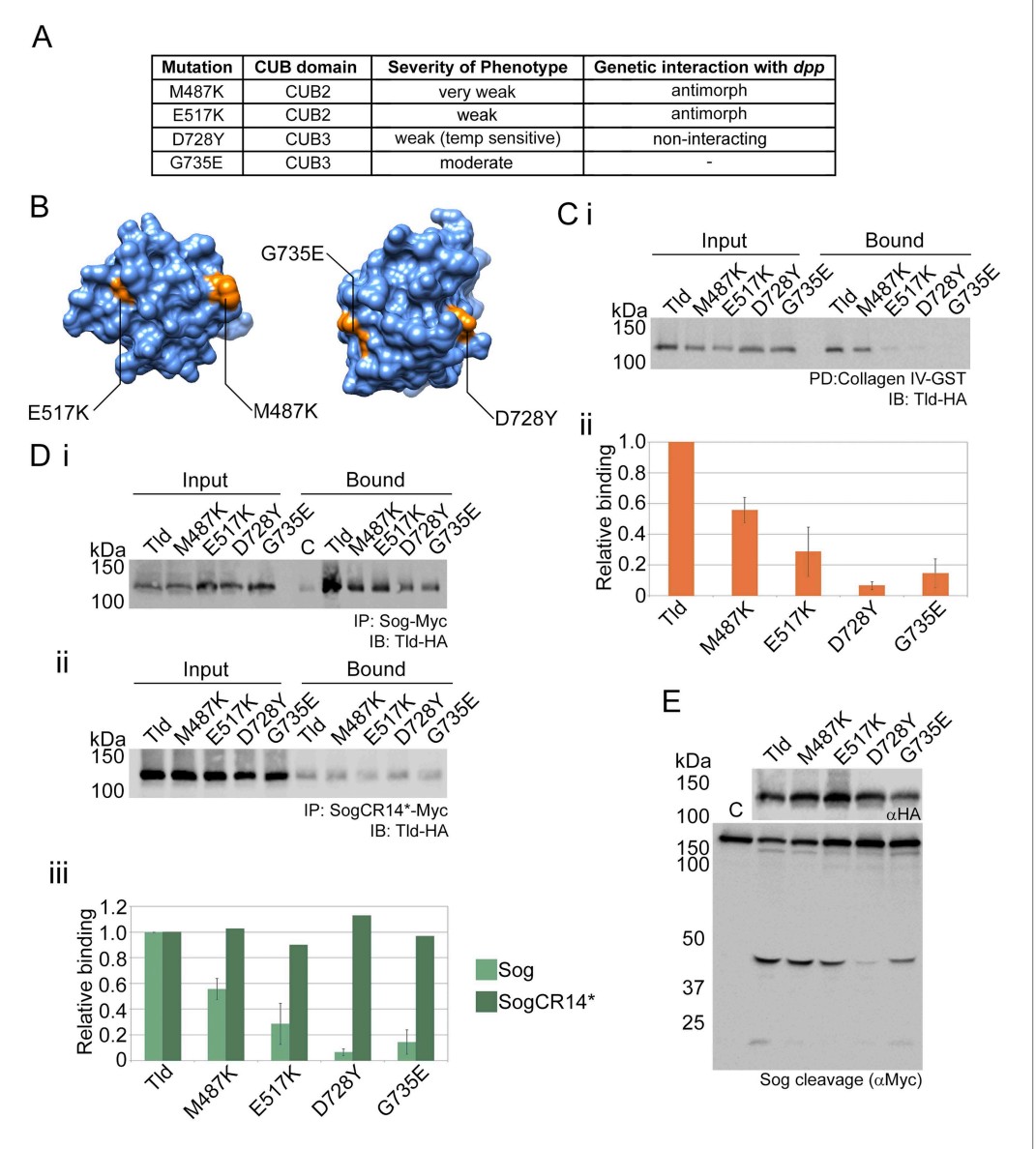

**Figure 5**. Point mutations affect Collagen IV binding. (**A**) Overview of previous data relating to classical *tld* alleles carrying point mutations in the CUB2 and CUB3 domains detailing the single amino acid change, extent of ventralisation and genetic interaction with *dpp* (**Childs and O'Connor 1994**; **Finelli et al. 1994**). (**B**) Models of Tld CUB2 and CUB3 with the single amino acid mutations highlighted in orange. (**Ci**) Western blot showing binding of the Tld-HA proteins to GST-VkgNC1, in the presence of Dpp, relative to the input levels. (**Cii**) Graph showing quantification of binding to Collagen IV normalised to the input levels, relative to binding of full-length Tld. Error bars are SEM, n = 3. (**D**) Western blot with anti-HA showing input levels of the catalytically inactive Tld-HA proteins tested and the amounts bound to Sog-Myc (**Di**) and SogCR14*-Myc (**Dii**) in immunoprecipitation experiments. (**Diii**) Graph showing the level of binding to Sog (light green) and SogCR14* (dark green) relative to that for full-length Tld in each case. Error bars for Sog are SEM, n = 3. (**E**) Western blot (top, anti-HA) showing Tld-HA proteins with the indicated single amino acid changes. Western blot (lower, anti-Myc) showing Sog cleavage assays (4 hr) using the Tld enzyme indicated above each lane in the presence of Dpp.

fragments plus the additional zipper sequence. Using this artificial system, Sog and the Tld MP domain can interact in the absence of Dpp, although Dpp still enhances SogZ+ cleavage (data not shown).

As the MPZ− and SogZ+ protein system facilitated interaction and SogZ+ cleavage, we tested the ability of these proteins to rescue Dpp gradient formation in a *tld* null background. To this end,

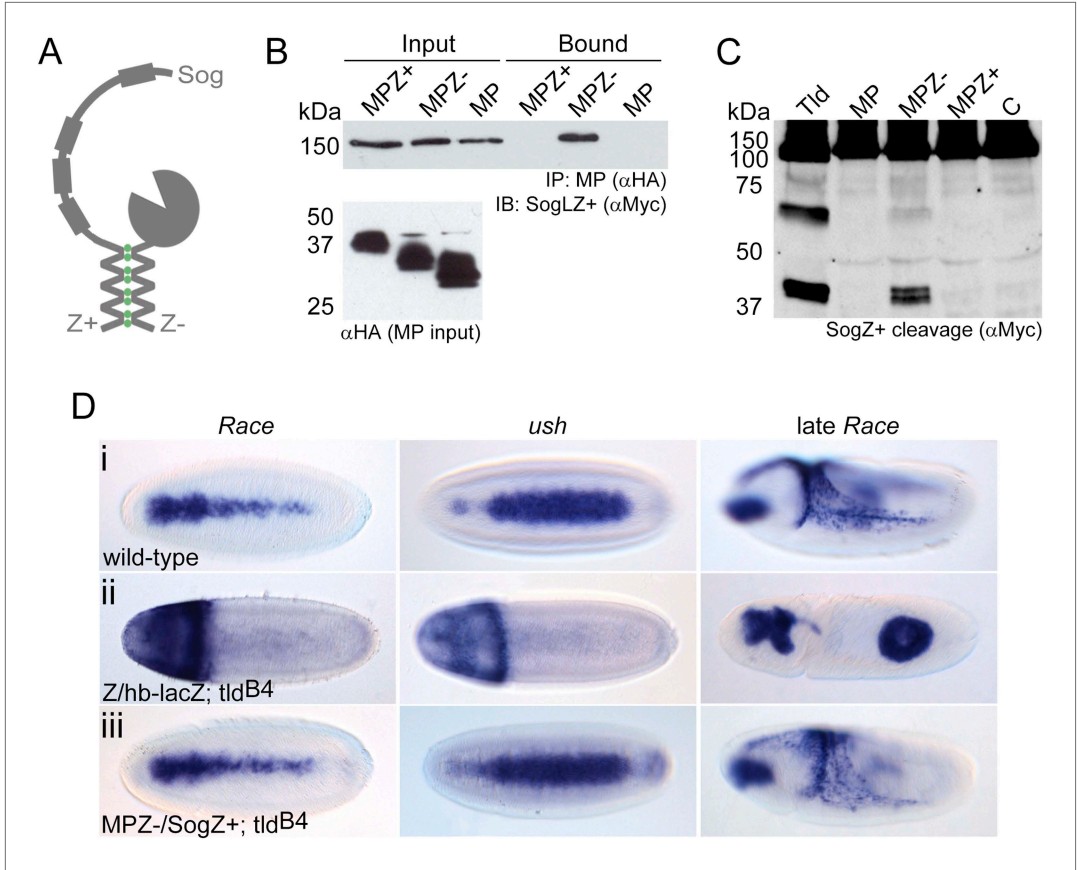

**Figure 6**. Restoration of in vivo function to the Tld MP domain. (**A**) Cartoon of the leucine zipper strategy. (**B**) Western blot (anti-HA) showing the input levels of the MP-HA and MP-Zipper-HA fusion proteins tested, and binding of Sog-Zipper+-Myc following immunoprecipitation with anti-HA. MPZ+ is larger than MPZ− due to the presence of additional glycine residues preceding the zipper sequence. (**C**) Western blot (anti-Myc) showing cleavage assays carried out with SogZ+ transfected with the indicated Tld MP-zipper fusion proteins in the presence of Dpp. C = control assay without Tld added. (**D**) Dorsal views of stage 6 embryos, oriented with anterior to the left, stained by RNA in situ hybridisation for the Dpp target genes *Race* and *u-shaped,* and stage 9 lateral views of late *Race* expression. Wild-type embryos are shown for reference in (**Di**). Embryos in (**Dii**) and (**Diii**) are also stained with a *lacZ* RNA probe that marks the chromosome carrying wildtype *tld* (*ftz-lacZ*, stripes, not shown) or chromosome lacking the zipper transgene (*hb-lacZ*, anterior staining). All embryos shown are *tld* mutant, and either lack at least one zipper transgene (**Dii**), or carry both zipper transgenes (**Diii**). See also *Figure 6—figure supplement 1*.

The following figure supplement is available for figure 6:

**Figure supplement 1**. SogZ + overexpression results in a thinning of *Race* expression.

SogZ+ was introduced into embryos as a transgene under the control of the *sog* lateral stripe enhancer (*Markstein et al., 2002*; *Peluso et al., 2011*), whereas the MPZ− domain was expressed under *tld* regulatory sequences (*Kirov et al., 1994*) which are sufficient to rescue the null embryonic phenotype when used to drive expression of wildtype *tld* (data not shown). In a *tld* null background with expression of either SogZ+ or MPZ− but not both, *Race* and *ush* expression are lost (*Figure 6Dii*, cf *Figure 6Di*). In contrast, when *tld* null mutant embryos express both SogZ+ and MPZ−, target gene expression is rescued in 50% of cases (*Figure 6Diii*), although ~20% of the rescued embryos show a slight thinning of *Race* expression in the central region of the embryo, as observed for the addition of an extra copy of ls-*sogZ+* (*Figure 6—figure supplement 1*). Overall, these findings indicate that this artificial tethering of the MP domain to Sog can be sufficient for BMP gradient formation in vivo.

## Discussion

### A model for the Collagen IV enhancement of Tld activity

Our biophysical data provide evidence that Tld is a monomer, which efficiently cleaves Sog due to critical bipartite functions of the Tld non-catalytic domains in either substrate or ECM interaction. Thus, Tld represents the first five CUB domain family member that exists as a monomer, as Tll-1 and mTld have previously been shown to be dimers that exclude Chordin substrate resulting in low catalytic activity towards it (*Berry et al., 2009, 2010*). Thus, whereas the dimeric Tolloids require the affinity of substrate interaction to be higher than the intramolecular affinity within the dimer for activation (*Hintze et al., 2006*; *Berry et al., 2009*), this is not a limitation for *Drosophila* Tld. We show that the Tld CUB1-3 domains bind Collagen IV. Consistent with this, recently an interaction has been identified between Collagen IV and the CUB domains of PCPE-1 (*Salza et al., 2014*), and Collagen IV binds the zebrafish G-protein coupled receptor GPR126, via a region containing the CUB domain (*Paavola et al., 2014*). We also show that Tsg enhancement of Sog processing by Tld is dependent on the Tld-Collagen IV interaction. Finally, we identify a strategy for artificially tethering the Tld MP domain to Sog in vivo, which can rescue BMP gradient formation.

Based on our findings, we revise the model of BMP gradient formation in the *Drosophila* early embryo (*Holley et al., 1996*; *Marques et al., 1997*; *Ashe and Levine, 1999*; *Sawala et al., 2012*), as shown in *Figure 7*. In addition to Sog and Dpp-Scw, which have been previously shown to bind independently to Collagen IV (*Sawala et al., 2012*), the data presented here show that Tld, via its CUB1-3 domains, also binds to Collagen IV (*Figure 7*, step 1). In the model, we suggest that initially there is no interaction between Sog, Dpp-Scw or Tld bound to Collagen IV. However, in the next step, when Dpp-Scw is transferred to Sog from Collagen IV, as has been previously described (*Sawala et al., 2012*), we suggest that Tld is transferred at the same time, interacting with Sog via the Tld CUB4 and CUB5 domains (*Figure 7*, step 2). Tsg has previously been shown to release Dpp-Scw/Sog from Collagen IV as a Dpp-Scw/Tsg complex (*Wang et al., 2008*), with Tsg recruitment or interaction potentially stabilised by the BMP binding capacity of Tsg's N-terminal domain (*Oelgeschlager et al., 2000*). Therefore, based on the data obtained here, we suggest that in addition Tld is released from Collagen IV at the same time, pre-bound to the shuttling complex (*Figure 7*, step 3), and is then able to fully process Sog (step 4).

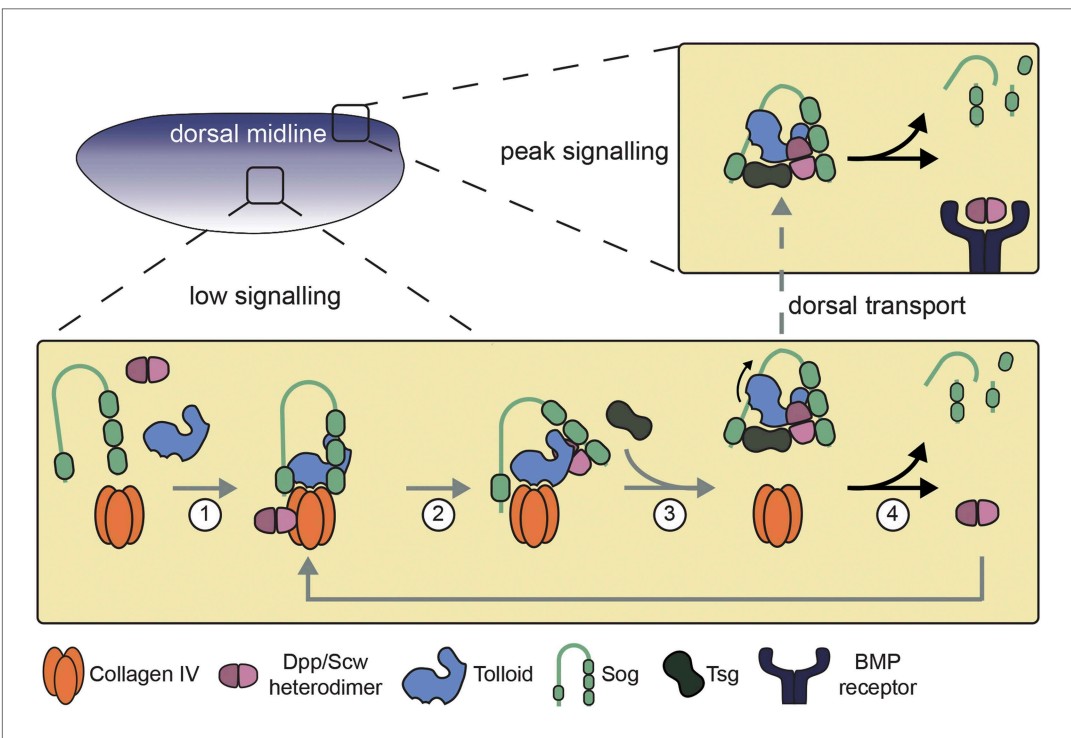

**Figure 7**. Model of Dpp gradient formation showing the Tld-ECM interaction. See text for details.

The model predicts that the enhancement of Tld cleavage of Sog that we observe in the presence of Collagen IV is due to Collagen IV acting as a scaffold to promote Tld interaction with Sog. We show reduced Collagen IV enhancement of Sog processing in vitro in the presence of high Tsg levels, relative to that detected with lower Tsg levels. As Tsg has been suggested previously to compete with Collagen IV for binding to the SogCR1 domain to release the Sog-BMP complex from Collagen IV (*Sawala et al., 2012*), it is possible that there is little Sog binding to Collagen IV when Tsg is in excess, so the positive effect of Collagen IV is reduced.

In the model presented, the BMP dependency for Sog cleavage by Tld arises from the requirement for Dpp-Scw to disrupt the SogCR4-Collagen IV interaction as described previously (*Sawala et al., 2012*), resulting in an 'opening' of Sog that facilitates Tld binding. In support of this, we find that Tld is unable to bind to Sog in the absence of Dpp (data not shown). In addition, our mechanism of gradient formation puts the emphasis on a balance between the distance that the Tld-BMP-Sog-Tsg complex diffuses and the rate with which Tld cleavage occurs, rather than on a limitation in terms of Tld recruitment. Although Tld levels are not limiting in the *Drosophila* embryo (*Eldar et al., 2002*), it is possible that Tld activity is suboptimal in order to allow sufficient diffusion required for peak gradient formation. Consistent with this idea, mathematical modelling has revealed that lowering Tld processing activity favours a more defined BMP peak, due to an increase in BMP shuttling to the dorsal midline (*Peluso et al., 2011*). In addition, a recent model of BMP shuttling in the sea urchin embryo finds Tld to be a dominant determinant of the system, with reduced levels of Tld relative to the fly system necessary for the best fit to the spatial extent of the BMP signalling peak in sea urchins (*van Heijster et al., 2014*).

## Complete Tld processing of Sog and the role of Tsg

In the absence of Collagen IV, or the binding of Sog or Tld to it, we observe little Tsg enhancement of Sog cleavage and liberation of the 33 kDa fragment. For example, all of the CUB2/3 point mutants are more severely compromised in their ability to cleave the 33 kDa site, compared to the 50 kDa site. Similarly, we also show that Tld cleavage of a mutant form of Sog, which is defective in Collagen IV interaction, does not release the 33 kDa fragment. As Tsg releases the BMP-Sog-Tld complex from Collagen IV in our model, we suggest that this release involves a remodelling of protein interactions that alters the interaction of Tld with Sog allowing better access to all Sog cleavage sites, but especially the 33 kDa site, which is Tsg-dependent (*Shimmi and O'Connor, 2003*). Moreover, we speculate that Tld can cleave the 50 kDa Sog site, that does not require Tsg, when the BMP-Sog-Tld complex is bound to Collagen IV, although the dual binding of Tld to Collagen IV and Sog holds Tld in a conformation that prevents both processing at the 33 kDa site and inactivation of Sog, prior to Tsg release. If these assumptions are correct, it follows that cleavage at the 33 kDa site within Sog is necessary for complete loss of Sog inhibition of Dpp-Scw. In support of this, the Sog C-terminal fragments generated by cleavage at the other two sites retain some BMP inhibitory activity when assayed in the *Drosophila* wing (*Yu et al., 2004*). In addition, complete cleavage of Chordin, the mammalian Sog ortholog, is required for relief of BMP inhibition (*Troilo et al., 2014*). We also note that although our zipper system can rescue BMP signalling thresholds in the embryo, cleavage at the 33 kDa equivalent cleavage fragment of the Sog-zipper fusion protein is the preferred site, consistent with cleavage at this site being key to full release of the Sog inhibitory effect.

In addition, if in *tsg* mutants Tld is unable to completely release the BMP inhibition of Sog, this may explain the only partial rescue of *tsg* mutants by overexpression of Supersog (*Yu et al., 2000*). Supersog consists mainly of the Sog CR1 domain and has been suggested, like Tsg, to compete with full-length Sog for binding to Collagen IV to release the shuttling complex (*Sawala et al., 2012*). However, it may be that incomplete processing of Sog by Tld in the absence of Tsg does not relieve full inhibition of the BMP ligands. Such a role for Tsg would be in addition to the Sog-independent positive role with respect to BMP signalling that has also been described (*Wang and Ferguson, 2005*).

## Tld domain roles and classical *tld* mutant phenotypes

The hypothesised roles of *Drosophila* Tld CUB domains in both Collagen IV and Sog interaction provide molecular insight into the phenotypes of classical *tld* alleles arising from point mutations in these domains. Mutations in CUB1-2 and CUB4-5 reduce Collagen IV and Sog binding, respectively, resulting in lower levels of Sog cleavage. As the shuttle-transport mechanism relies on cycles of complex diffusion, destruction and reformation, decreased Sog cleavage disrupts formation of the Dpp activity gradient resulting in ventralisation. Our in vitro data show that the CUB2 deletion behaves differently from the

CUB2 point mutants in that, although the ΔCUB2 and point mutants all show reduced binding to Collagen IV, unlike the point mutants, the ΔCUB2 mutant remains able to bind Sog, although it cleaves very poorly. We suggest that the ΔCUB2 mutant is conformationally different so it can bind to Sog without prior presentation by Collagen IV, but cleavage is affected, especially of the 33 kDa fragment, and not enhanced by Tsg, which may reflect perturbation of the Tld–Sog interaction, as proposed above.

The CUB2 point mutant *tld* alleles have been shown to be antimorphic when transheterozygous with either weak *tld* or *dpp* alleles (*Ferguson and Anderson, 1992*; *Finelli et al., 1994*). Revertants of these antimorphic mutations have also been identified within the CUB1, 2 and 5 domains (*Childs and O'Connor, 1994*), but as they reduce secretion of the mutant Tld proteins (*Lee et al., 2009*), they provide only limited insight into the molecular mechanism of the antimorphic mutants. Nevertheless, based on the finding that *Drosophila* and vertebrate Tld CUB1/2 or CUB4/5 domain pairs can bind BMP4 and BMP7, and inhibit BMP signalling when injected into *Xenopus* embryos (with reciprocal inhibition of Tld by BMP interaction), it was proposed that the antimorphic effect of *tld* alleles may be due to an inhibition of Dpp activity (*Lee et al., 2009*). However, the role of the Tld-BMP interaction in the *Drosophila* embryo is not fully understood and only point mutants in the CUB2 or protease domains behave in an antimorphic manner (*Ferguson and Anderson, 1992*; *Finelli et al., 1994*). One possibility is that Tld and Dpp binding to Collagen IV largely prevents a direct Tld–Dpp interaction, but instead favours shuttling complex assembly in the presence of Sog and Tsg. In the absence of Collagen IV binding, the CUB2 point mutants could inhibit Dpp to some extent, further reducing Dpp activity that is already compromised in a background heterozygous for a weak *tld* or *dpp* allele.

### Relevance to other Tolloid family members

The differential specificity of Tolloid non-catalytic domains, and in particular ECM interaction, may be relevant to the fine tuning of Tolloid activity not only during BMP gradient formation in other organisms, but also in a wide range of developmental contexts in general. In *Xenopus* embryos, the Chordin/BMP ligand gradient diffuses within the Fibronectin-rich Brachet's cleft (*Plouhinec et al., 2013*), and Fibronectin has also been shown to bind to Chordin and BMP1, to enhance the processing of Chordin during DV patterning of vertebrate embryos (*Huang et al., 2009*). Similarly, ONT1, a secreted Olfactomedin-class protein, facilitates Chordin-BMP1 interaction by independently recruiting both proteins, resulting in robust BMP gradient formation during embryonic DV patterning. In addition, for procollagen III processing by BMP1 in the presence of PCPE-1, heparin-like sulphated glycosaminoglycans may act as a scaffold for BMP1 processing of procollagen III in vivo (*Bekhouche et al., 2010*). Therefore, Tld family members may generally rely on a scaffold to facilitate substrate interaction, as proposed here for Collagen IV.

Using a leucine zipper based interaction system (*Luan et al., 2006*) we were able to show that artificially tethering the Tld MP domain to Sog is sufficient to support BMP gradient formation. It would be interesting to determine whether tethering the BMP1 protease domain to Chordin can circumvent the requirement for the BMP1 non-catalytic domains in vertebrates. Perhaps this would allow too much turnover of Chordin in vivo given that BMP1 cleavage of Chordin is independent of BMPs. In addition, BMP1 MP domain tethering would prevent interaction with secreted Frizzled-related proteins, which are recruited via the non-catalytic domains of mammalian Tolloids to mediate important further positive and negative control of activity in specific regions of vertebrate embryos (*Muraoka et al., 2006*; *Kobayashi et al., 2009*; *Ploper et al., 2011*; *De Robertis and Colozza, 2013*; *Inomata et al., 2013*). It is interesting that *Drosophila* lacks secreted Frizzled-related proteins, suggesting that ECM control of the Tld-substrate interaction is sufficient for fine control of Tld activity and the resulting Sog and BMP levels in the *Drosophila* embryo. This may stem from Tld processing of Sog being BMP-dependent in *Drosophila*, a situation that has already been shown to favour the steep BMP gradient required in this context (*Peluso et al., 2011*). Given that the leucine zipper system mediates an interaction between two different proteins synthesised in distinct regions of the embryo, we suggest that it will be useful for not only determining sufficiency of other interactions, but also for determining the effect of mediating ectopic or prolonged interactions between other test proteins in vivo.

## Materials and methods

### DNA plasmids

The Cu-inducible SogCR14*-Myc (*Sawala et al., 2012*) and Sog-Myc plasmids and pGEX4T1-VkgC have been described (*Wang et al., 2008*). The Cu-inducible Tsg-V5 plasmid was generated by

inserting the Tsg CDS from pRmHa1-Tsg-His (*Yu et al., 2000*) into pMT-V5-His. Tld-HA, in pRmHA1-aTld (*Marques et al., 1997*), was modified by PCR to introduce deletions and single amino acid mutations. To generate VkgC-FLAG and Dcg1C-FLAG the NC1 domain was inserted into pMTBip followed by a C-terminal FLAG tag. Leucine zipper constructs were generated by insertion of zipper sequences from pActPL-Gal4AD and pActPL-Gal4DBD (*Luan et al., 2006*) into Sog-Myc and MP-HA. For injection into *Drosophila* embryos the CDS from MPLZ- was inserted into pCasper-attB containing the *tld* promoter as defined previously (*Kirov et al., 1994*), *tld* 3'UTR and 1 kb of additionally regulatory sequences. The CDS of Sog-Myc, SogCR14*-Myc or SogLZ+-Myc was inserted into pCasper-attB, together with the Sog promoter and 3'UTR taken from pCasper-ls-Sog (*Peluso et al., 2011*). For large-scale protein expression Tld was catalytically inactivated by mutation E94A using Quikchange mutagenesis (Agilent Technologies, Santa Clara, CA) and ligated into pCEP-PU using a Not1 restriction enzyme to incorporate a 6 × His tag at the C-terminus.

## Protein expression and purification

Tld-HA, Sog-Myc, Tsg-V5, VkgC-FLAG and Dcg1C-FLAG were produced in *Drosophila* S2R+ cells using the Effectene transfection kit (Qiagen, Venlo, Netherlands) and 2 µg DNA per well in a six-well plate format. Protein expression was induced after 24 hr by the addition of 500 µM $CuSO_4$ and secreted proteins harvested after 72 hr. For leucine zipper cleavage assay transfections, S2R+ cells were co-transfected with a combination of 1.5 µg SogLZ+-Myc, and 0.5 µg MPZ+/MPZ−/MP or Tld as a positive control. For RNAi-mediated knock down of Collagen IV, cells were transfected and incubated for 24 hr at 25°C. The cells were then treated for 1 hr in serum-free GIBCO media with 5 µg *viking* dsRNA (generated from *yw* genomic DNA using MEGAscript T7 with the primers listed below), followed by the addition of FBS and Cu-induction 24 hr later.

5'-TAATACGACTCACTATAGGGGTCCAATAGCTCCTTGCTCG-3'
5'-TAATACGACTCACTATAGGGATCCGATGGTAGCAAAGGTG-3'

For Tld expression used in the biophysical analysis, HEK293-EBNA cells were maintained at 37°C under 5% CO2 in DMEM/F12 10% FBS, 0.1 unit/ml penicillin, and 10 µg/ml streptomycin growth media. The Lipofectamine transfection reagent (Invitrogen, Carlsbad, CA) was used to generate HEK293-EBNA stable cell lines using pCep-PU-Tld$^{E94A}$. After 24 hr, selection was started with the addition of 5 µg/ml puromycin. The stable cell lines were then maintained using a lower level of selection at 1 µg/ml puromycin. Tld was harvested and purified using a HisTrap Ni-NTA column followed by size-exclusion chromatography using a Superdex200 column in the presence of 10 mM Tris HCl (pH7.4) with 0.5 M NaCl, 1 mM $CaCl_2$ and 2 M Urea. GST-VkgC and GST were expressed and purified as described (*Wang et al., 2008*).

## Small angle X-ray scattering

SAXS data for dTld were collected on EMBL Hamburg SAXS beamline P12 at Petra III. The scattering images obtained were averaged and buffer scattering intensities subtracted using PRIMUS software. The radius of gyration and distance distribution functions were calculated using GNOM (*Svergun, 1992*) and particle shapes were modelled using DAMMIN (*Svergun, 1999*). 10 models were combined to produce an average shape using DAMAVER (*Svergun, 1999*). Rigid-body modelling of dTld domains to the experimental scattering data was performed using SASREF (*Petoukhov and Svergun, 2005*). For use in rigid-body modelling Tld domains were modelled on solved crystal structures of similar domains using SWISS-MODEL (*Schwede et al., 2003*; *Arnold et al., 2006*; *Kiefer et al., 2009*). Templates included; the BMP1 protease domain (*Mac Sweeney et al., 2008*), human TSG6-CUBC and human MASP1/3 from the protein databank (*Berman et al., 2000*). During the modelling procedure each domain was separated by a 4 Å spacer and numerous runs performed to elucidate the most consistent structure.

## Transmission electron microscopy

Tld (10 µl at 34 µg/ml) was analysed by negative stain using 2% phosphotungstic acid at pH7.4. Data was collected on a FEI Tecnai Twin Transmission Electron Microscope at 120 kV. Images were recorded at 31,000× magnification between −0.5 and −1 µm defocus on a 2048 × 2048 pixel camera (Gatan Orius SC1000). The total number of particles in the dataset was 2297. Particles were selected manually using EMAN2 (*Ludtke et al., 1999*) with a 96 × 96 pixel window (30 × 30 nm). Imagic5 (*van Heel et al., 1996*) was used to align the particles and sort them into classes by multivariate statistical analysis. The classes were then used as references to realign the dataset through various rounds

of realignment. When particle classes appeared stable, euler angles were assigned to the images to generate a 3D reconstruction that was subject to further rounds of refinement.

## Cleavage assays

To perform cleavage assays the protein containing media from transient transfections was mixed and incubated for 1–7 hr at 25°C. Reaction volumes ranged from 30–51 μl containing: 10–30 μl Sog-Myc with 0–10 μl Tld-HA proteins and a combination of 0–10 μl Tsg-V5, 0–10 μl VkgC-FLAG/Dcg1C-FLAG (co-transfected) and 1–20 nM recombinant Dpp (rDpp). Recombinant Dpp was purchased from R&D systems (Minneapolis, MN) and reconstituted in 4 mM HCl. Total reaction volume was kept constant by the addition of mock transfected cell medium. When complete, reactions were stopped with sample buffer (2.5% glycerol, 12.5% SDS, 20% β-mercaptoethanol, 24 mM Tris/HCl) and analysed by Western blot. Cleavage assays using the zipper fusion proteins were performed by co-transfecting the appropriate plasmids, as described above.

Antibodies used as follows; 1:500 anti-Myc antibody (clone 4A6, Millipore, Darmstadt, Germany), 1:1000 anti-V5 tag (monoclonal AV5-PKI, AbCam, Cambridge, UK), 1:1000 anti-6xHis tag (AbCam), 1:1000 anti-HA (Roche, Basel, Switzerland), 1:1000 anti-flagM2 (SIGMA). Blots were visualised using either ECL reagent (GE Healthcare, Buckinghamshire, UK) or West Dura Supersignal (Thermo Scientific, Waltham, MA) with the ChemiDoc MP System (BioRad, Hercules, CA) or exposed using Biomax film (Kodak, Rochester, NY). For quantification ImageJ Version 10.2 or ImageLab (BioRad) was used.

## Sog immunoprecipitation and VkgC GST-Pulldown

Equivalent amounts of Tld-HA proteins were incubated with 300 μl Sog-Myc, or SogCR14*-Myc containing media with 30 μl 50% anti-Myc agarose matrix (Sigma, St. Louis, MO) prewashed in IP buffer (50 mM Tris pH7.5, 150 mM NaCl, 0.02% TritonX-100) in the presence of 1 nM Dpp for 2 hr at 4°C. Following binding, the beads were washed multiple times in IP buffer and resuspended and heated to 95°C for 5 min in 30 μl 1XSDS sample buffer for analysis by Western blot. The same principle was followed for leucine zipper immunoprecipitations in the absence of rDpp. GST pulldown assays were carried out using 10–300 μl Tld-HA proteins (levels normalised by Western blot) as described (*Sawala et al., 2012*).

## Fly stocks

Fly stocks used were *y*[67c23]*w*[118], *tld*[B4]/TM3, *tld*[2]/TM3 (Bloomington Stock Center, Bloomington, IN). Embryos were collected from *tld*[2]/TM3, and *yw* parents as well as crosses of *MPZ–/CyO, hb-lacZ*; *tld*[B4]/TM3, *ftz*-lacZ to *sogZ+/CyO, hb-lacZ*; *tld*[B4]/TM3, *ftz-lacZ*. Embryos were fixed and antibody stain or RNA in situ hybridisation with digU-labelled RNA probes was carried out using standard protocols. Width of the *ush* stripe was quantified using ImageJ.

## Acknowledgements

We thank the Bloomington Stock Centre for fly stocks, Benjamin White for the zipper DNAs, Mihaela Serpe for the ls-sog expression plasmid, Michael O'Connor for pRmHA1-aTld, Cath Sutcliffe for technical assistance and the staff in the EM facility and Chris Bayley for EM assistance (University of Manchester). Parts of this research were carried out at the light source Petra III at DESY, a member of the Helmholtz Association (HGF). We would like to thank Cy Jeffries for assistance in using beamline P12. This research was supported by a Wellcome Trust programme grant (092005/Z/10/Z) to HLA, a Wellcome Trust Studentship (083271/Z/07/Z) to AS, a BBSRC studentship to JW and a BBSRC grant (BB/I019286/1) to CB. The *Drosophila* Core Research Facility at the Faculty of Life Sciences, University of Manchester was established through funds from the University and the Wellcome Trust (087742/Z/08/Z). The Wellcome Trust Centre for Cell-Matrix Research, University of Manchester, is supported by core funding from the Wellcome Trust (088785/Z/09/Z).

## Additional information

### Funding

| Funder | Grant reference number | Author |
| --- | --- | --- |
| Wellcome Trust | 092005/Z/10/Z | Hilary L Ashe |

| Funder | Grant reference number | Author |
|---|---|---|
| Biotechnology and Biological Sciences Research Council | BB/I019286/1 | Clair Baldock |
| Biotechnology and Biological Sciences Research Council | | Jennifer Winstanley |
| Wellcome Trust | 083271/Z/07/Z | Annick Sawala |
| Wellcome Trust | 088785/Z/09/Z | Jennifer Winstanley, Clair Baldock, Hilary L Ashe |

The funders had no role in study design, data collection and interpretation, or the decision to submit the work for publication.

### Author contributions

JW, Conception and design, Acquisition of data, Analysis and interpretation of data, Drafting or revising the article; AS, Acquisition of data, Analysis and interpretation of data, Drafting or revising the article; CB, HLA, Conception and design, Analysis and interpretation of data, Drafting or revising the article

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
