## [Decision Letter]

Thank you for sending your work entitled “Synthetic enzyme-substrate tethering obviates the Tolloid-ECM interaction during *Drosophila* BMP gradient formation” for consideration at *eLife*. Your article has been favorably evaluated by K VijayRaghavan (Senior editor) and 3 reviewers, one of whom is a member of our Board of Reviewing Editors. 

The Reviewing editor and the other reviewers discussed their comments before we reached this decision, and the Reviewing editor has assembled the following comments to help you prepare a revised submission. 

There was a consensus that the data are of high quality, that the problem is important, and that the results are largely convincing. For these reasons, the reviewers believe this paper to be broadly suitable for publication in *eLife*. 

The concerns raised are mostly quite minor and/or matters of interpretation to be addressed in the text. Perhaps the two most substantial are the role of Dpp in promoting Sog cleavage, and the explanation of some of the classical mutants. 

Firstly, in *Drosophila*, in contrast to *Xenopus*, ligand-bound Sog is cleaved more efficiently than free Sog, and this feature is essential for establishing the Dpp activation gradient. When the experiments of Figure 2 are first presented, it is important to emphasize this point, and that Dpp is present in the in vitro assays. It would also be good to demonstrate in their assays that, in the absence of Dpp, the cleavage of Sog by Tld is indeed inefficient. This would lend more credibility to the biological relevance of the assay. 

Secondly, one of the unusual genetic aspects of certain *tld* alleles is that they behave antimorphically with respect to certain *dpp* alleles. The classical interpretation of these antimorphs is that they titrate Dpp into a non-functional complex. This makes sense for the *tld* antimorphic alleles that map to the protease domain since these might still be assembled via Col IV interaction into a non-cleavable complex but the CUB2 antimorphs are a bit more difficult to explain since these should not interact well with Col IV and might be expected not to allow complex formation. The authors appear to argue that for these antimorphic mutations, the complex does not form, and as a result Dpp will remain bound to collagen and thereby be hindered in its movement towards the dorsal most region. It is not clear how this fits with the behavior of the cis-acting revertants of the 10E 95 antimorphic mutations that where isolated by [11]. Some of these revertants mapped to CUBs1 and2 while others where in CUB5 (8). These all might also be expected to leave Dpp bound to collagen, yet they revert the antimorphic behavior of the 10E95 allele. Some comment about these alleles with respect to the model seems warranted. 

Other issues:

1) When discussing the possibility of limiting levels of Tld in vivo, the 2002 Eldar et al. paper (which is also authored by Ashe) should be mentioned, where Tld heterozygous backgrounds gave rise to a wt pattern of pMad.

2) The paper mentions the difference between monomeric *Drosophila* Tld and the dimeric form of Tld in other species, which excludes the substrate. It would be interesting to speculate in the Discussion on how dimeric Tld molecules may be activated and induced to associate with their substrate.

3) In discussing the (quite complex) overall model, it would be good to make explicit for which aspects there is good evidence, and which elements are more speculative. It is important for a non-specialist reader to be clear about this distinction.

---

## [Author Response]

*Firstly, in* Drosophila*, in contrast to* Xenopus*, ligand-bound Sog is cleaved more efficiently than free Sog, and this feature is essential for establishing the Dpp activation gradient. When the experiments of*
Figure 2
*are first presented, it is important to emphasize this point, and that Dpp is present in the in vitro assays. It would also be good to demonstrate in their assays that, in the absence of Dpp, the cleavage of Sog by Tld is indeed inefficient. This would lend more credibility to the biological relevance of the assay*.* *

We have emphasized this point in the second paragraph of the Results section and added a figure (Figure 2—figure supplement 1) showing that Sog cleavage by Tld is Dpp-dependent.

*Secondly, one of the unusual genetic aspects of certain* tld *alleles is that they behave antimorphically with respect to certain* dpp *alleles. The classical interpretation of these antimorphs is that they titrate Dpp into a non-functional complex. This makes sense for the* tld *antimorphic alleles that map to the protease domain since these might still be assembled via Col IV interaction into a non-cleavable complex but the CUB2 antimorphs are a bit more difficult to explain since these should not interact well with Col IV and might be expected not to allow complex formation. The authors appear to argue that for these antimorphic mutations, the complex does not form, and as a result Dpp will remain bound to collagen and thereby be hindered in its movement towards the dorsal most region. It is not clear how this fits with the behavior of the cis-acting revertants of the 10E 95 antimorphic mutations that where isolated by*
[11]*. Some of these revertants mapped to CUBs1 and2 while others where in CUB5 (*[8]*). These all might also be expected to leave Dpp bound to collagen, yet they revert the antimorphic behavior of the 10E95 allele. Some comment about these alleles with respect to the model seems warranted. *

There is actually a rather trivial explanation for the behavior of the revertant mutations, which is that they have all been shown previously to either reduce or prevent Tld secretion ([22], Genes Dev 23: 2551). As such, they eliminate the antimorphic function of the Tld mutants that is mediated extracellularly. We now mention this in the discussion of the antimorphic mutations (in the Discussion section).

*Other issues*:

*1) When discussing the possibility of limiting levels of Tld in vivo, the 2002 Eldar et al. paper (which is also authored by Ashe) should be mentioned, where Tld heterozygous backgrounds gave rise to a wt pattern of pMad*.

This paper is now cited in the Discussion section.

*2) The paper mentions the difference between monomeric* Drosophila *Tld and the dimeric form of Tld in other species, which excludes the substrate. It would be interesting to speculate in the Discussion on how dimeric Tld molecules may be activated and induced to associate with their substrate*.

We have inserted a comment in the beginning of the Discussion section, speculating on how the dimer is activated.

*3) In discussing the (quite complex) overall model, it would be good to make explicit for which aspects there is good evidence, and which elements are more speculative. It is important for a non-specialist reader to be clear about this distinction*.* *

We have modified the text explaining the model (first subsection in the Discussion) and now either make it clearer when we are referring to aspects of our data for which there is good evidence or cite the appropriate references for other steps where there are supporting data. For the elements that are more speculative, we highlight these by introducing them with wording such as ‘we suggest’, ‘the model predicts’, etc. We have also divided what was previously one single paragraph relating to the model into two separate paragraphs, so that the first paragraph focuses on more concrete aspects of the model, whereas the following two paragraphs are more speculative.